# *EuPIP1;2*, a Plasma Membrane Aquaporin Gene from *Eucommia ulmoides*, Enhances Drought and Salt Tolerance in Transgenic Tobacco

Jiajia Chen, Jianrong Li, Yanhui Huang, Yan Li [ID], Changfeng Su and Xiaofang Zeng *

Key Laboratory of Plant Resource Conservation and Germplasm Innovation in Mountainous Region (Ministry of Education), Institute of Agro-Bioengineering, College of Life Sciences, Guizhou University, Guiyang 550025, China; chenjj220@163.com (J.C.); jrli@gzu.edu.cn (J.L.); hangyanhui999@163.com (Y.H.); yanli@gzu.edu.cn (Y.L.); scf1984911278@163.com (C.S.)
* Correspondence: xfzeng1@gzu.edu.cn

**Abstract:** Aquaporins are a specific type of membrane channel proteins that efficiently transport water molecules and other small molecular substrates in plants. In this study, we isolated the plasma membrane aquaporin gene *EuPIP1;2* from *Eucommia ulmoides*, an important medicinal plant in China. The EuPIP1;2 protein was localized on the plasma membrane. Quantitative RT-PCR analysis revealed that *EuPIP1;2* was constitutively expressed in all analyzed tissues, with the highest expression levels detected in the fruit and root. Overexpression of *EuPIP1;2* in transgenic tobacco enhanced plant tolerance of drought and salinity. Under drought and salt stress, the transgenic lines exhibited higher percentage germination, longer roots, and enhanced percentage survival compared with wild-type plants. The contents of malonaldehyde and proline suggested that *EuPIP1;2* improved drought and salt tolerance in transgenic lines by reducing damage to membranes and improving osmotic adjustment. We predict that *EuPIP1;2* could be applied to improve drought and salt tolerance in transgenic plants.

**Keywords:** *EuPIP1;2*; drought stress; salt stress; resistance

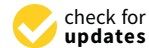



## 1. Introduction

Water is essential for plant growth and development, and water deficit can limit plant growth and even cause plant death due to water imbalance [1]. To adapt to environmental stresses, plants often activate specific physiological and biochemical mechanisms, such as the formation of various proteins to resist damage caused by water shortage [2]. Aquaporins (AQPs) are a specific type of membrane channel protein that efficiently transports water molecules and other small molecular substrates in plants [3]. AQPs are membrane-localized proteins classified within the major intrinsic protein (MIP) superfamily [4]. Since the isolation of the first member of the plant tonoplast intrinsic protein (TIP) subfamily from *Arabidopsis thaliana*, AQPs have been identified in *Arabidopsis thaliana* [5], *Zea mays* [6], *Oryza sativa* [7], *Glycine max* [8], *Solanum lycopersicum* [9], *Gossypium hirsutum* [10], and *Brassica rapa* [11]. Based on phylogenetic and subcellular localization analyses, plant AQPs constitute five subfamilies: TIP, plasma membrane intrinsic proteins (PIP), NOD26-like intrinsic proteins (NIP), small and basic intrinsic proteins (SIP), and unidentified X intrinsic proteins (XIP) [4,12].

In plants, AQPs form pores in cell membranes and control the flow of water molecules in and out of cells by osmotic potential, thus regulating the water balance [3]. Under water-deficiency, plants can dynamically stabilize water content by adjusting the abundance and activity of AQPs, improve water use efficiency, and enhance tolerance to external stress [2]. Therefore, AQPs play an important role in maintaining water balance in plant cells. *PIPs* are the largest subfamily of plant AQPs and play an integral role in water transport

between cells [13]. *PIPs* are located on the plasma membrane as homo- or heterologous tetramers, and form highly selective channels that facilitate the passage of water, $CO_2$, and other small solutes across cell membranes [14]. Thus, *PIPs* play an important role in seed germination, cell elongation and differentiation, stomata and leaf movement, and fruit ripening [12]. Overexpression of *PIP* genes may improve resistance to osmotic stress in diverse plant species, such as Arabidopsis [15], rice [16], chrysanthemum [17], banana [18], and tomato [19].

With global drought and water shortage, and intensification of soil salinization, the functions of plant AQPs and the response mechanism to drought stress have become a major research focus. There are large differences in expression, stress response patterns, and functions of *AQP* gene in different plants, tissues, and cells. Therefore, studying the functions of aquaporins in different plants during plant growth and development and stress resistance is of great significance to further understand and systematically elaborate the physiological functions and mechanisms of aquaporins under normal and stress conditions. *Eucommia ulmoides* is an important medicinal plant in China [20]. The function of the *AQP* gene has been reported in several species. However, the function of *AQP* genes in *Eucommia ulmoides* remains unclear. Research on the function of AQPs and the drought-stress response mechanism in *E. ulmoides* has provided insights into the function and stress response mechanism of AQPs in general. In the current study, a plasma membrane *AQP* gene, *EuPIP1;2*, from *E. ulmoides* was identified. Assessment of stress resistance indicated that overexpression of *EuPIP1;2* improved the drought and salt tolerance of transgenic tobacco. The present results are of important theoretical value and provide a novel genetic resource for the improvement of stress resistance in forest trees.

## 2. Materials and Methods

### 2.1. Plant Materials

Ten-year-old female plants of *Eucommia ulmoides* were selected for gene cloning and expression pattern experiments. Leaves of *Nicotiana benthamiana* were used as the recipient for visualization of the subcellular localization of the protein. Tobacco (*Nicotiana tobacum*) "K326" was used as the wild-type (WT) control and all transgenic lines were generated in the background of "K326" in this study. All transgenic and WT plants were grown in a glasshouse (16 h light/8 h dark cycles; 24 °C; relative humidity 70%).

### 2.2. Gene Cloning and Bioinformatic Analysis of EuPIP1;2

Total RNA was extracted from leaves for gene cloning using TRIzol Reagent. Purified RNA (1 μg) was reverse-transcribed into cDNA using the RNA PCR Kit (TaKaRa, Kusatsu, Japan) in accordance with the manufacturer's instructions. Primers were designed based on the full-length coding sequence to isolate and clone EuPIP1;2 (EUC16674) from *E. ulmoides* [21]. A neighbor-joining tree was constructed based on a multiple alignment of *EuPIP1;2* and *PIPs* from other plant species using DNAMAN 6.0.3.99 and MEGA 11 software. The protein's three-dimensional structure and homology was modeled using the SWISS-MODEL server (https://swissmodel.expasy.org/ (accessed on 16 February 2022)).

### 2.3. Quantitative RT-PCR and RT-PCR Analysis

Total RNA was extracted from leaves, roots, stems, and fruit for gene expression analysis using TRIzol Reagent. Reverse transcription and quantitative RT-PCR (qRT-PCR) analysis were performed as described by Zeng and Zhao [22]. The *EuActin* gene was used as an internal reference [23]. Amplifications were conducted using the qRT-PCR system (Bio-Rad, Hercules, CA, USA). The relative gene expression level was calculated from three independent samples using the $2^{-\Delta\Delta Ct}$ method [24]. For the RT-PCR analysis, the total RNA was isolated from tobacco by using the RNA-prep pure Plant Kit (Omega total RNA Kit I, Norcross, GA, USA). 5 μg RNA was used as a template for reverse transcription (RT) according to the Primer Script® RT-PCR Kit protocol (Genstar, Beijing, China). The *Actin* gene (NM_001325686.1) was used as an internal reference [25]. RT-PCR following the

method of Zeng and Zhao [26]. The primers used for qRT-PCR and RT-PCR analysis are listed in Table 1.

**Table 1.** Primers used in the study.

| Primers | Primer Sequences | Purpose |
|---|---|---|
| *EuPIP1;2C*-F | 5′-GGGAGTTTCTAGAACGAACAACA-3′ | Gene cloning |
| *EuPIP1;2C*-R | 5′-TCAAGAGGAGCTCTTAAATGGGA-3′ | |
| *EuPIP1;2*-F | 5′-TGAACCACGGCTACACCA-3′ | qRT-PCR and RT-PCR |
| *EuPIP1;2*-R | 5′-AACCCAATCGGAAGAGGC-3′ | |
| *EuActin*-F | 5′-TGACGGAGCGTGGTTACTCATTCA-3′ | qRT-PCR |
| *EuActin*-R | 5′-TCTTGGCAGTCTCCATTTCCTGGT-3′ | |
| *EuPIP1;2J*-F | 5′-TGAACCACGGCTACACCA-3′ | Transgenic plant PCR detection |
| *EuPIP1;2J*-R | 5′-AAGGGACCCACCCAGAAA-3′ | |
| *NtActin*-F | 5′-TGGTTAAGGCTGGATTTGCT-3′ | RT-PCR |
| *NtActin*-R | 5′-TGCATCCTTTTGACCCATAC-3′ | |

### 2.4. Vector Construction and Plant Transformation

For overexpression vector construction, the full-length coding sequence of *EuPIP1;2* was amplified by chemical synthesis and inserted into the pSH737 vector, which contained the *Cytomegalovirus* 35S promoter and a nopaline synthase terminator (Figure 1). The vector was introduced into *Agrobacterium tumefaciens* strain EHA105, and then transformed into tobacco "K326" using the *Agrobacterium*-mediated method in accordance with the protocols of Lu et al. [27]. Transgenic plants were confirmed using a β-glucuronidase (GUS) histochemical assay in accordance with Jefferson et al. and PCR detection [28]. The primers used for transgenic plant detection are listed in Table 1.

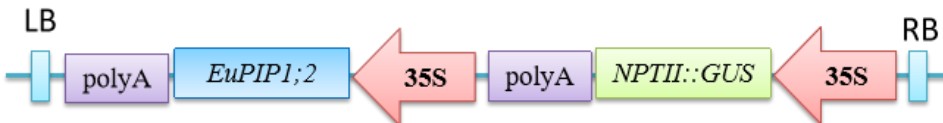

**Figure 1.** Schematic illustration of the pSH-*35S-EuPIP1;2* expression vector.

### 2.5. Protein Subcellular Localization

To determine the subcellular localization of *EuPIP1;2*, the pCambia-35S-*EuPIP1;2*::GFP vector was constructed. *Agrobacterium tumefaciens* strain EHA105 harboring this vector was introduced into *N. benthamiana* leaves for transient expression following the method of Li, et al. [29]. The *A. tumefaciens* strain EHA105 harboring the pCambia-35S::GFP vector was used as a control. The green fluorescent protein (GFP) fluorescence signal was observed with a laser confocal microscope (Leica TCS SP8 STED, Wetzlar, Germany) under excitation at 488 nm.

### 2.6. Drought Treatment

For the germination test, $T_2$ seeds from the transgenic lines P1 and P4 as well as WT seeds were sown on filter paper moistened with 0 or 150 mM mannitol and incubated for 7 days at 24 °C under 16 h light/8 h dark cycles. For the root length and hypocotyl assays, seeds were treated with 75% ethanol for 30 s, then sterilized with 10% ($v/w$) NaClO solution for 10 min, and rinsed with sterile water five times. The seeds were cultured on half-strength Murashige and Skoog (1/2 MS) medium supplemented with 0 or 150 mM mannitol for 15 days at 24 °C under 16 h light/8 h dark cycles, then the root and hypocotyl lengths were measured. For drought stress treatment, water was withheld from 25-day-old tobacco seedlings in pots filled with soil (Klasmann-Deilmann TS1 876, Germany) for 30 days in a glasshouse, and then the plants were watered as normal for 7 days.

### 2.7. Determination of Water Loss

Four-week-old seedlings of the WT and the transgenic lines ($T_2$ generation) P1 and P4 were used for water loss analysis following the method described by Aharoni et al. [30]. The shoot was detached from the roots and weighed immediately to record the fresh weight. All shoot samples were placed on open petri dishes at room temperature (22 °C, relative humidity 45%) and were weighed at 0, 20, 30, 60, 120, 180, 240, 300, 360, and 420 min after excision to record the desiccated weight (DW). Measurements were recorded from five individual plants of the WT and each transgenic line. The percentage water loss was calculated with the formula: water loss rate (%) = (FW − DW)/FW × 100.

### 2.8. Salt Treatment

For the germination test, $T_2$ seeds from the transgenic lines P1 and P4 as well as WT seeds were sown on filter paper moistened with 0 or 100 mM NaCl and incubated for 7 days at 24 °C under 16 h light/8 h dark cycles. For the root length assay, seeds were treated with 75% ethanol for 30 s, then sterilized with 10% (*v/w*) NaClO solution for 10 min, and rinsed with sterile water five times. The seeds were cultured on 1/2 MS medium for 7 days, then transferred to MS medium supplemented with 0 or 150 mM NaCl for 20 days at 24 °C under 16 h light/8 h dark cycles, and then the root length was measured. For salt stress treatment, 25-day-old plants in pots filled with soil (Klasmann-Deilmann TS1 876, Geeste, Germany) were watered with 300 mM NaCl for 15 days in a glasshouse.

### 2.9. Determination of Malondialdehyde and Proline Contents

Twenty-five-day-old seedlings of the WT and the transgenic lines P1 and P4 were transferred to MS medium supplemented with 0 or 150 mM mannitol or 150 mM NaCl for 7 days, then leaves were collected for determination of malondialdehyde (MDA) and proline (PRO) contents. The contents were measured using commercial detection kits (Suzhou Keming, Suzhou, China) in accordance with the manufacturer's instructions.

### 2.10. Statistical Analysis

The data are presented as means ± SD. The significance of differences among the means was analyzed with one-way ANOVA followed by Duncan's multiple range test at the significance level of α = 0.05, using IBM SPSS Statistics 25 software.

## 3. Results

### 3.1. Cloning and Characterization of EuPIP1;2

The cDNA of EuPIP1;2 was isolated from *E. ulmoides*. Sequence analysis revealed that the EuPIP1;2 open reading frame was 858 bp and encoded 286 amino acids. EuPIP1;2 contained six typical transmembrane domains, two highly conserved "NPA" (Asn-Pro-Ala) motifs of the MIP protein family (Figure 2a, orange box), the MIP family intrinsic protein-specific sequence (Figure 2a, green box), and two plant PIP conserved sequences (Figure 2a, purple box). Phylogenetic analysis indicated that EuPIP1;2 was most closely related to CsPIP1;2 (*Camellia sinensis*) and most distantly related to PIP proteins of the monocotyledonous species rice and maize (Figure 2b). Protein homology-modeling predicted that EuPIP1;2 formed a homologous tetramer (Figure 2c).

### 3.2. Expression Pattern of EuPIP1;2

The expression pattern of *EuPIP1;2* in the root, stem, leaf, and fruit of *E. ulmoides* was investigated. The qRT-PCR analysis demonstrated that *EuPIP1;2* transcripts were constitutively expressed in all of the analyzed organs, among which the relative expression level was highest in the fruit and root, followed by the stem and leaf (Figure 3a). The RT-PCR results showed that the *EuPIP1;2* had been expressed in transgenic tobacco (Figure 3b).

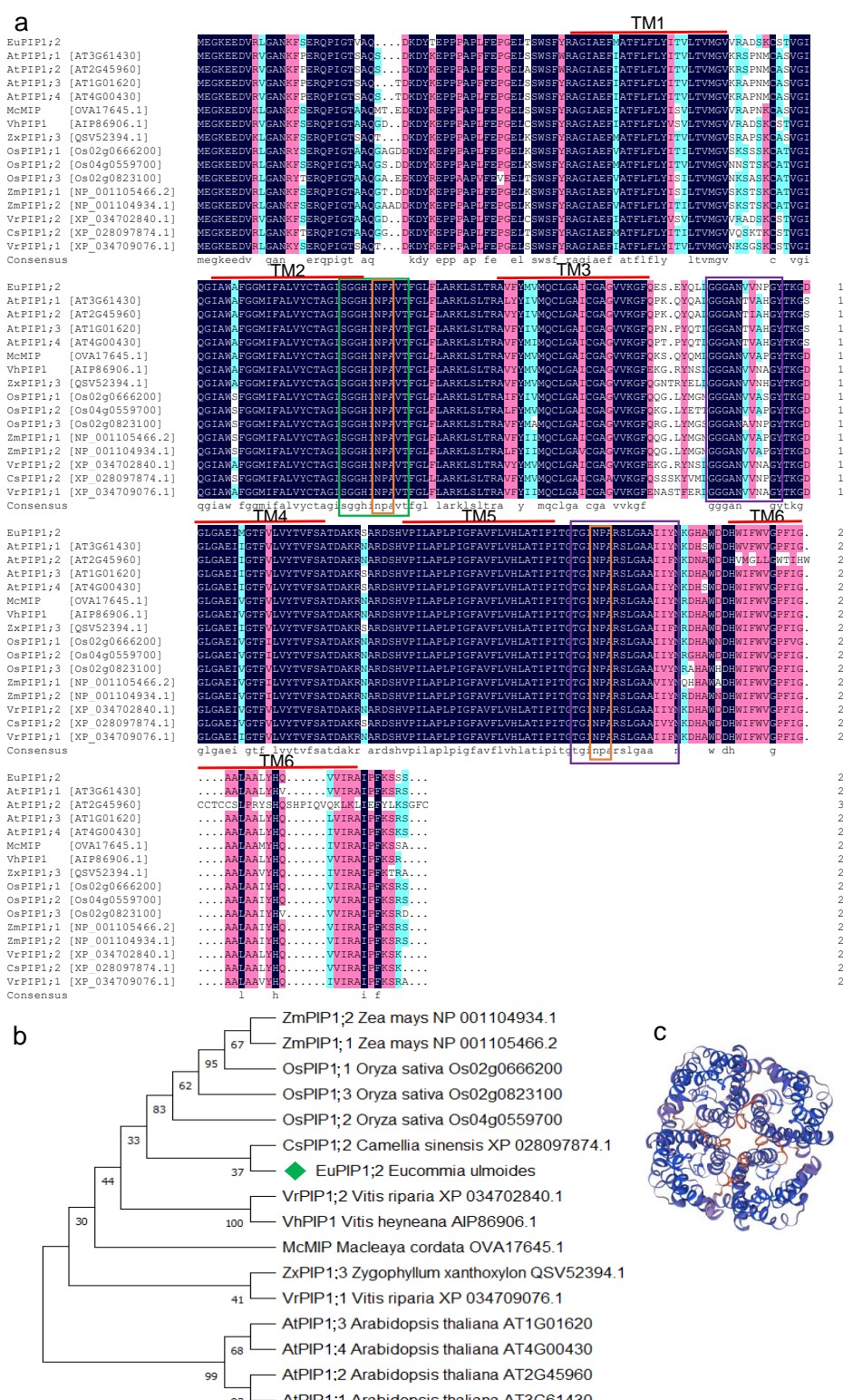

**Figure 2.** Multiple alignment and phylogenetic analysis of EuPIP1;2 with other plant PIP proteins. (**a**) The conserved amino acid residues in all proteins. The MIP family intrinsic protein-specific motif "SGGHINPAVT" is enclosed in the green box, the plant PIP highly conserved sequences "GGGANVVNPGY" and "TGINPARSLGAAIIYN" are indicated by purple boxes, and two "NPA" motifs are enclosed in orange boxes. TM, transmembrane domain. (**b**) Phylogenetic analysis of EuPIP1;2 and PIP proteins from other plant species. (**c**) Prediction of the tertiary structure of EuPIP1;2.

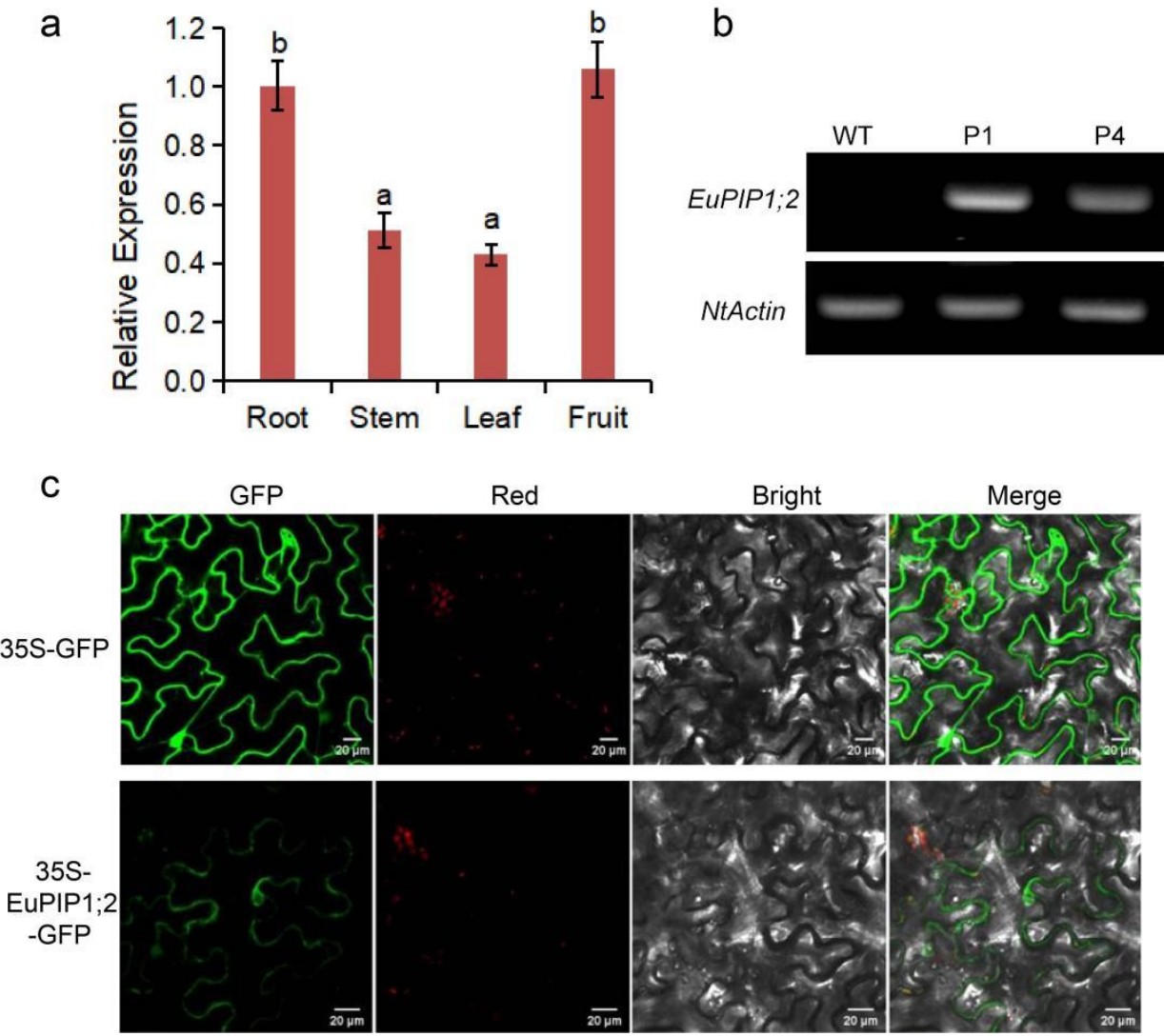

**Figure 3.** (**a**) Relative expression level of *EuPIP1;2* in four organs of *Eucommia ulmoides*. (**b**) Expression analysis of the *EuPIP1;2* and *NtActin* in wild-type (WT) tobacco and transgenic lines. (**c**) Subcellular localization of *EuPIP1;2* in cells of *Nicotiana benthamiana* leaves.

### 3.3. Subcellular Localization of EuPIP1;2 Protein

To determine the subcellular localization of EuPIP1;2, the 35S-EuPIP1;2-GFP construct was transiently expressed in *N. benthamiana* leaves. The EuPIP1;2 protein was localized on the plasma membrane, which further indicated that the protein belonged to the PIP subfamily (Figure 3c).

### 3.4. Overexpression of EuPIP1;2 Improved Drought Tolerance of Transgenic Tobacco

To evaluate the function of *EuPIP1;2*, an overexpression vector harboring *EuPIP1;2* was transformed into tobacco "K326" for heterologous expression. Twenty transgenic individuals were identified by GUS histochemistry and PCR detection (Figure A1). Two $T_2$ transgenic lines P1 and P4 were selected to explore the effect of *EuPIP1;2* on drought and salt tolerance. Wild-type and transgenic tobacco seeds were germinated on filter paper moistened with 0 or 150 mM mannitol for 7 days. Mannitol treatment reduced the percentage germination of WT, P1, and P4 seeds. The percentage germination of P1 and P4 seeds was significantly higher than that of WT seeds (Figure 4a,b).

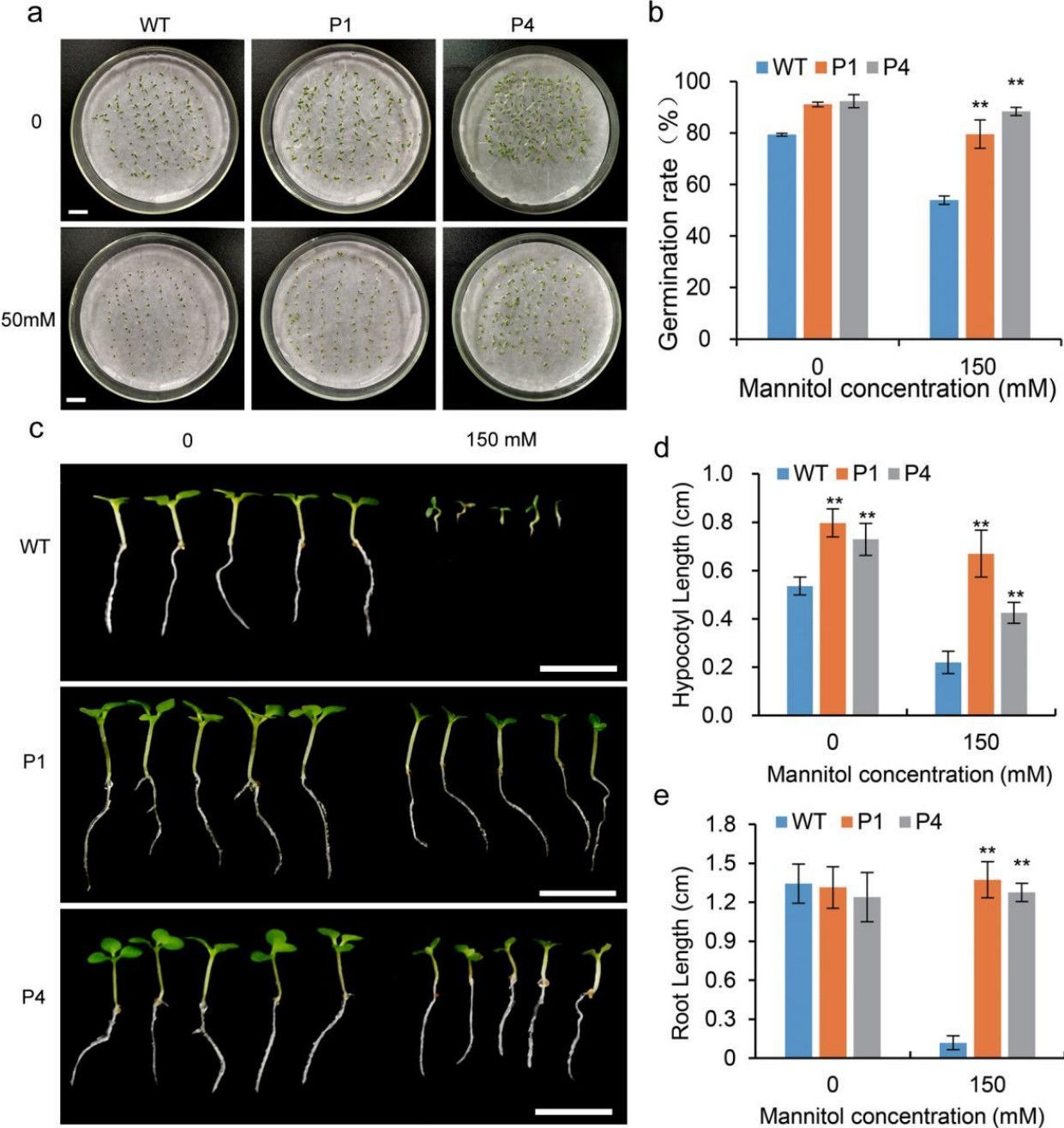

**Figure 4.** Transgenic lines (P1 and P4) and wild-type (WT) tobacco "K326" seedlings treated with 0 or 150 mM mannitol. (**a**) Germination of seeds (bar = 1 cm), (**b**) percentage germination of seeds, (**c**) phenotype of the seedlings after 15 days (bar = 1 cm), (**d**) hypocotyl length, and (**e**) root length. Data were analyzed statistically using SPSS. ** $p < 0.01$.

To assess the effect of *EuPIP1;2* on the drought resistance of transgenic tobacco seedlings, seeds of the two transgenic lines and WT were grown on a medium supplemented with 0 or 150 mM mannitol, and the lengths of the root and hypocotyl were measured after 15 days. In the absence of mannitol, the root length of transgenic seedlings did not differ significantly from that of WT seedlings, whereas the hypocotyl of transgenic seedlings was significantly longer than that of WT seedlings; the hypocotyl length of the P1 and P4 lines was 1.48 and 1.36 times that of the WT, respectively (Figure 4c–e). Under drought stress, the roots of WT seedlings essentially did not elongate (0.19 cm), whereas the mean root length of P1 and P4 seedlings was 1.37 cm and 1.27 cm, respectively (Figure 4c–e).

The hypocotyl length of the P1 and P4 transgenic seedlings was significantly longer (by 3.04 and 1.93 times, respectively) compared with that of WT seedlings (Figure 4).

To further study the drought resistance of transgenic tobacco overexpressing *EuPIP1;2*, water was withheld from 25-day-old WT and transgenic tobacco lines (P1 and P4) for 30 days and then watered as normal for 7 days. The survival of transgenic tobacco plants was higher than that of the WT (Figure 5a). To determine the effect of *EuPIP1;2* on water loss in vitro of tobacco seedlings, the rate of water loss by the shoot of WT and transgenic plants was measured. The rate of water loss of P1 and P4 plants was lower than that of the WT tobacco, and the difference attained significance after 300 min (Figure 5b). These results indicated that overexpression of *EuPIP1;2* improved the drought resistance of transgenic tobacco, which may be associated with a reduction in the rate of leaf water loss.

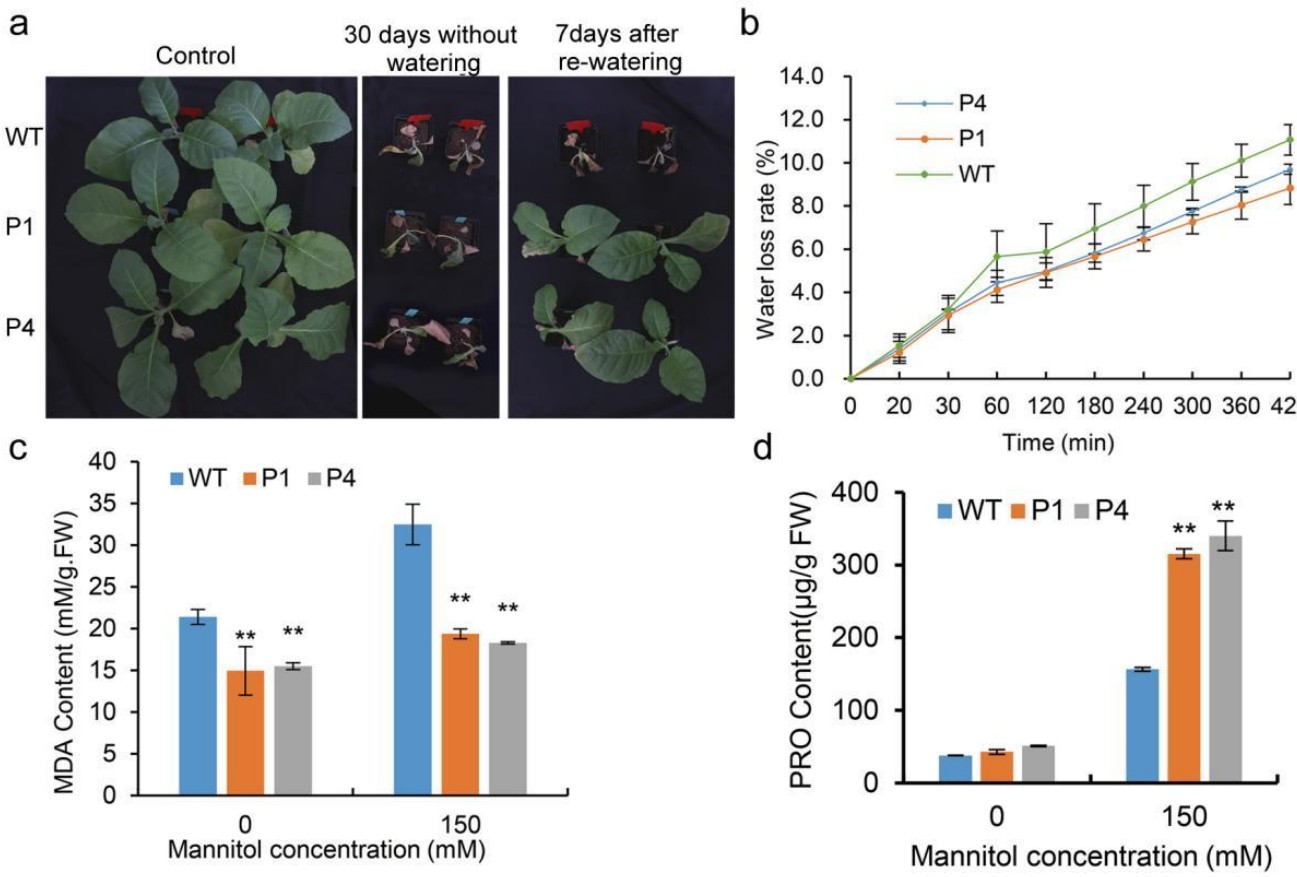

**Figure 5.** Drought resistance of wild-type (WT) and *EuPIP1;2*-overexpressing tobacco. (**a**) The WT and *EuPIP1;2*-overexpressing tobacco plants after drought stress treatment for 30 days and rewatering for 7 days. (**b**) Water loss from the leaves of WT and *EuPIP1;2*-overexpressing tobacco in vitro. (**c**) Malondialdehyde (MDA) content in leaves of WT and *EuPIP1;2*-overexpressing tobacco treated with 0 or 150 mM mannitol. (**d**) Proline content in leaves of WT and *EuPIP1;2*-overexpressing tobacco treated with 0 or 150 mM mannitol. Data were analyzed statistically using SPSS. ** $p < 0.01$.

The content of MDA reflects the degree of cell membrane damage in plants, and PRO content can be used as a biochemical indicator of plant resistance. To further study the effects of *EuPIP1;2* overexpression on the physiology and biochemistry of transgenic tobacco, the MDA and PRO contents in leaves of transgenic and WT plants under mannitol treatment were measured. Under the non-stress treatment, the MDA contents of P1 and P4 transgenic plants were significantly lower (by 69.77% and 72.34%, respectively) compared with that of the WT. Under drought stress treatment, the MDA content of WT plants increased significantly, attaining 32.47 mM/g·FW, an increase of 11.07 mM/g· FW; in contrast, the MDA contents of P1 and P4 plants were 19.36 mM/g FW and 18.26 mM/g·FW,

which represented an increase only by 4.43 mM/g·FW and 2.76 mM/g· FW, respectively (Figure 5c). No significant difference in PRO content between WT and transgenic plants was observed under the non-stress treatment. However, under the drought stress treatment, the PRO contents of P1 and P4 plants were significantly higher (by 2.01 and 2.17 times, respectively) than that of the WT (Figure 5d). Therefore, we speculated that the increased drought resistance of transgenic plants compared with that of the WT might be associated with the low degree of membrane damage, rapid osmotic adjustment, and solute accumulation in transgenic plants.

### 3.5. Effects on Salt Tolerance of Transgenic Tobacco

To study the effect of *EuPIP1;2* on the salt tolerance of transgenic tobacco, seeds of WT and transgenic tobacco lines (P1 and P4) were germinated on filter paper moistened with 0 or 100 mM NaCl for 7 days. Under the non-stress treatment, no significant difference in percentage germination of WT and transgenic tobacco seeds was observed, whereas under 100 mM NaCl treatment, the germination frequency of transgenic tobacco seeds was significantly higher than that of the WT tobacco. The percentage germination of P1 seeds was 81%, whereas that of P4 seeds was almost unaffected (94%), and the percentage germination of WT seeds was 67% (Figure 6a,b). To further assess the salt resistance of transgenic tobacco seedlings, 7-day-old seedlings (P1, P4, and WT) were transferred to MS medium supplemented with 0 or 150 mM NaCl, and root length was measured after 20 days. Under the salt treatment, the root length of P1 and P4 seedlings was significantly longer (by 2.45 and 2.10 times, respectively) than that of the WT (Figure 6c,d). These results indicated that overexpression of *EuPIP1;2* improved the salt tolerance of tobacco seedlings.

To further evaluate the effect of *EuPIP1;2* on the salt tolerance of transgenic tobacco, 25-day-old WT, P1, and P4 tobacco seedlings in pots were watered with 300 mM NaCl for 15 days. Under the salt treatment, leaf growth of both transgenic and WT plants was inhibited, and a portion of the leaves of WT plants began to yellow and wilt, whereas the leaves of transgenic lines remained dark green and turgid (Figure 6e). This result verified that overexpression of *EuPIP1;2* improved the salt tolerance of tobacco. No significant difference in leaf PRO content was observed between WT and transgenic plants under the non-stress treatment. However, in response to the salt treatment, the PRO content in leaves of both transgenic lines and WT plants was increased; the PRO content of P1 and P4 plants was significantly higher (by 2.07 and 1.58 times, respectively) compared with that of the WT (Figure 6f). These results indicated that the increase in salt tolerance of transgenic tobacco might be associated with the enhanced osmotic adjustment and solute accumulation by transgenic plants under salt stress compared with that by the WT.

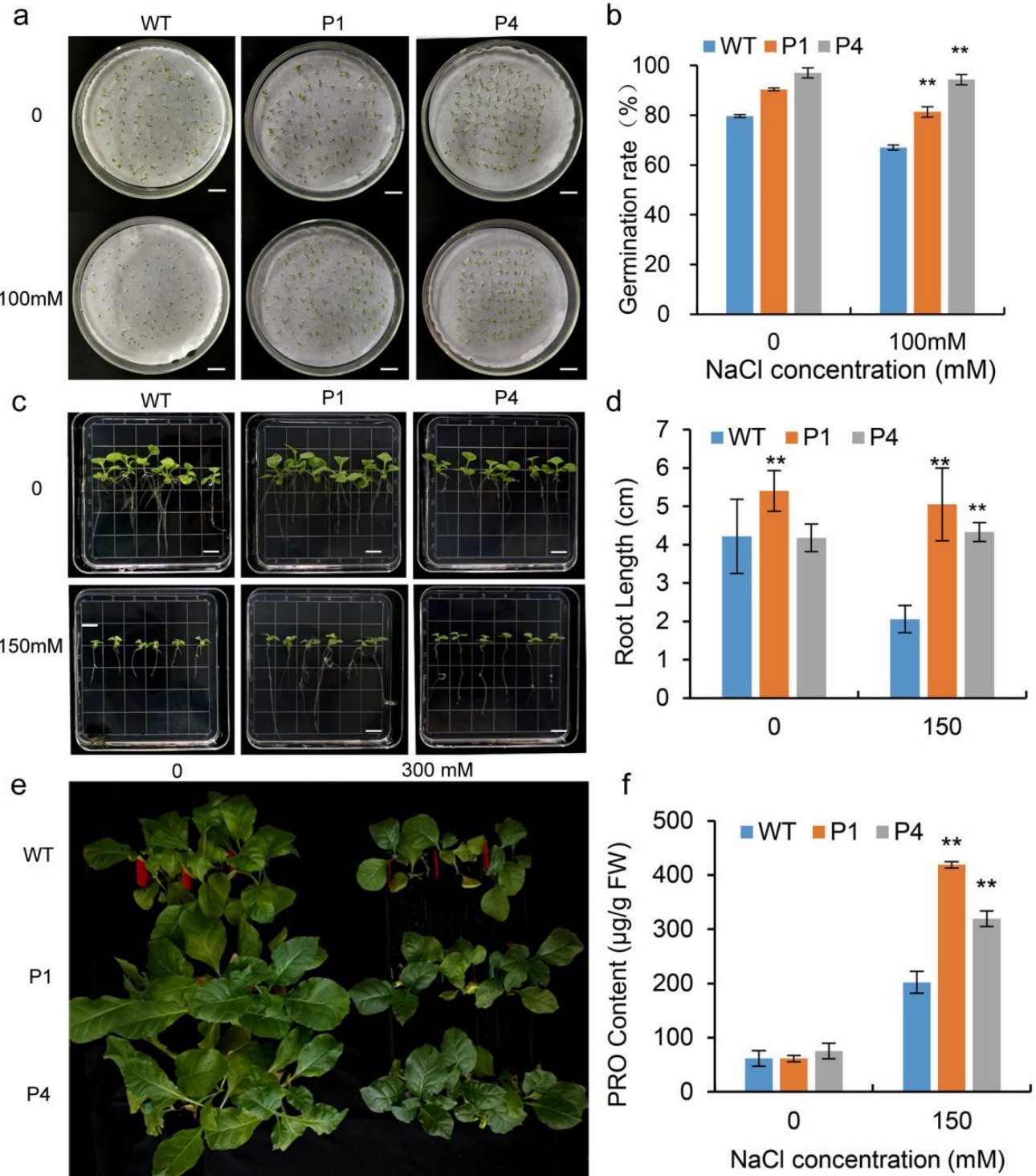

**Figure 6.** Seed germination and growth of wild-type (WT) and *EuPIP1;2*-overexpressing tobacco lines (P1 and P4) treated with NaCl. (**a**) Germination of seeds (bar = 1 cm) and (**b**) percentage germination of seeds treated with 0 or 100 mM NaCl. (**c**) Growth of seedlings (bar = 1 cm) and (**d**) root length of seedlings treated with 0 or 150 mM NaCl. (**e**) Phenotypes of WT and transgenic lines treated with 0 or 300 mM NaCL. (**f**) Proline (PRO) content in the leaf of seedlings treated with 0 or 150 mM NaCl. Data were analyzed statistically using SPSS. ** $p < 0.01$.

## 4. Discussion

Plasma membrane intrinsic proteins, the largest subfamily of plant MIPs, represent an integral pathway for intercellular water transport [31]. The EuPIP1;2 protein is localized on the plasma membrane. Analysis of conserved domains indicated that EuPIP1;2 is a member of the PIP1 group and is most closely related to *PIPs* of dicotyledonous woody plants and only distantly related to *PIPs* of monocotyledonous plants. Previous studies have shown that *PIP* genes are expressed in diverse tissues of plants [32], such as roots, leaves, stems, inflorescences, fruits, and seeds [33]. In the present study, the expression level of *EuPIP1;2* was highest in the fruit, followed by the root. The high expression level in the fruit may be associated with the rapid development of *E. ulmoides* fruit in August. The expression level of *EuPIP1;2* in the root was higher than that in the stem and leaf, which might indicate that *EuPIP1;2* was mainly involved in water transport in the roots.

Under water-limitation, *PIPs* increase root water conductivity, release solutes for osmotic adjustment, reduce water potential in plant cells, and further limit water loss by reducing transpiration, thereby enhancing plant resistance to drought [34]. Overexpression of *PIP* genes improves the drought resistance of transgenic plants [35]. The present research revealed that the rate of water loss of transgenic tobacco in vitro was significantly lower than that of the WT. Under drought stress, the percentage germination and root length of transgenic seedlings were higher than those of the WT, indicating that overexpression of *EuPIP1;2* improved the drought resistance of transgenic tobacco. In addition, overexpression of *PIP* genes improves the salt resistance of plants. For example, overexpression of *ScPIP1* [35], *ZxPIP1;3* [29], *CfPIP1-1*, *CfPIP1-2*, and *CfPIP1-4* [36], and *MdPIP1;3* [37] improve the salt and drought resistance of transgenic Arabidopsis. Similarly, the current study also showed that *EuPIP1;2* improved the salt resistance of transgenic tobacco.

In plants, reactive oxygen species (ROS) play a crucial role in the acclimation of plants to abiotic stress [38]. A high concentration of ROS can cause oxidative damage to membranes (by lipid peroxidation) and can lead to oxidative destruction of the cell in a process termed oxidative stress [39]. Therefore, as a product of membrane lipid peroxidation, the content of MDA reflects the degree of cell membrane damage in plants. Previous studies have shown that *PIP* genes reduce the MDA content and membrane damage under stress in transgenic plants, and assist transgenic plants to maintain a relatively healthy physiological state under stress, thereby improving stress resistance. Under drought or salt stress, transgenic plants that overexpress *ScPIP* [35], *ZxPIP1;3* [29], *MSPIP2;2* [32], *CmPIP1* [17], or *ScPIP2–1* [40] have a lower MDA content than WT plants. The present study showed that under drought stress, the MDA content of transgenic plants was significantly lower than that of WT plants, indicating that the membrane damage suffered by *EuPIP1;2*-overexpressing plants was not as severe as that experienced by WT type. Our results were consistent with previous findings that indicate *PIP* genes reduce membrane damage under different stress types [41].

In vivo, PRO not only regulates the permeability of the cytoplasm, but also stabilizes the structure of biological macromolecules, reduces cell acidity, eliminates ammonia toxicity, and regulates cell redox potential [42]. Accumulation of PRO in plants is associated with plant resistance to stress, thus, PRO content can be used as a biochemical indicator of plant resistance [43]. We observed that the PRO content of *EuPIP1;2*-overexpressing tobacco was significantly increased under drought and salt stress. These results showed that the accumulation of osmoregulatory substances in transgenic plants was significantly higher than that in the WT, which improved drought resistance and salt resistance of tobacco.

## 5. Conclusions

In this study, we cloned an aquaporin gene, *EuPIP1;2*, which encodes a protein localized on the plasma membrane in *E. ulmoides*. The expression level of *EuPIP1;2* was highest in the fruit, followed by the root, and lowest in the leaf. Under drought and salt stress, *EuPIP1;2* improved the resistance of transgenic tobacco by promoting root elongation,

reducing leaf water loss, reducing membrane damage, and increasing the content of proline in the leaf.

**Author Contributions:** J.C., J.L., Y.H., Y.L. and C.S., assisted in the experiments; J.C. and X.Z., wrote the manuscript; X.Z., supervised the research. All authors have read and agreed to the published version of the manuscript.

**Funding:** This research was funded by the Science and technology project in Guizhou Province, grant number (2017)5788.

**Institutional Review Board Statement:** Not applicable.

**Informed Consent Statement:** Not applicable.

**Data Availability Statement:** Not applicable.

**Conflicts of Interest:** The authors declare no conflict of interest.

**Appendix A**

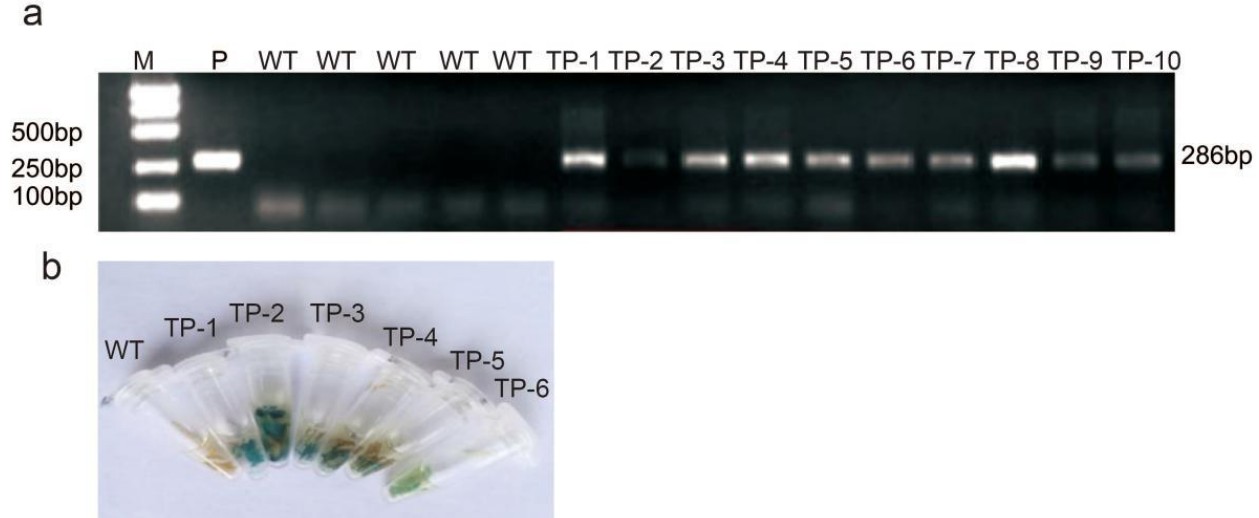

**Figure A1.** (**a**) PCR detection M: Marker DL2000 (TaKaRa, Japan) (**b**) GUS histochemical staining of transgenic tobacco.

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
