# Peer review of "EuPIP1;2, a Plasma Membrane Aquaporin Gene from Eucommia ulmoides, Enhances Drought and Salt Tolerance in Transgenic Tobacco"

_agronomy, doi:10.3390/agronomy12030615_

Round 1

Reviewer 1 Report

Summary:
The present study characterized an aquaporin gene (EuPIP1;2) identified in Eucommia ulmoides in response to drought and salt stress. EuPIP1;2 has plasma membrane domains, and its transcripts were abundant in 4 different tissue types analyzed. The authors checked the cellular localization of EuPIP1;2-GFP proteins in tobacco, identifying the GFP signal in the plasma membrane. The authors generated transgenic tobacco plants, which showed hyposensitivity to drought and salt stress compared to WT. In a drought condition, MDA contents significantly decreased while PRO contents increased in the transgenic lines relative to WT. Similarly, in a salt stress condition, PRO contents increased in the transgenic lines relative to WT. 

Major issues:
1) Figure 5: It looks like P1 and P4 had similar levels of MDA content between non-stress (0 mM Mannitol) and stress (150 mM Mannitol) conditions. Does that mean P1 and P4 did not have any cell membrane damage in the stress condition?
2) Line 168-169 (Figure 3a): To me, the expression level in root was similar to one in fruit.  
3) Figure 3b: GFP alone (35S:GFP) also localized in the plasma membrane. Is that normal?
4) Figure 4a: The germination assay was done in T2 lines of P1 and P4, which means that they were segregation pools. I expected to see a level of variations in the germination rate with 150 mM in P1 and P4. However, the variation was unexpectedly low. Especially, P4 has almost no variation. Why is that?
5) Figure 4: The authors should provide expression data of EuPIP1;2 gene in the tobacco transgenic lines to make sure it was truly overexpressed. 
6) Figure 5c: MDA contents were significantly low in 0 mM Mannitol in P1 and P4 relative to WT. Does that mean that the effects of EuPIP1;2 overexpression were not specific to the stress condition? 
7) Line 258-259: Fig 6e shows the opposite of what the authors described.

Minor issues:
1) Line 54: Add reference(s). 
2) Table 1:  Add gene ID to the table. 
3) Line 119: What generation were P1 and P4 lines?
4) Figure 2a: I would suggest labeling the transmembrane domain in the multialign figure. 
4) Figure 2b: Provide the species name from which each homolog gene came.
5) Line 274-276: Additional references are needed, as the authors mentioned “many previous studies”. 
2) Line 297 and 311: The authors introduced what MDA and PRO indicate here. Good, but I wonder if it would be better to add this information to the result section (e.g., before explaining the results of Figure 5c,d). That would help readers from different backgrounds better understand why they were checked.

Author Response

Dear reviewer,

Thank you very much for your advice. We are truly grateful to your critical comments. We feel lucky that the valuable comments helped us with the improvement of our manuscript.

Based on the comments we received, careful modifications have been made to the R1 manuscript. Please see the attachment, you will find our point-by-point responses to your comments.

I look forward to hearing from you soon.

With best wishes,

Yours sincerely,

Xiaofang Zeng

Reviewer 2 Report

The study is well performed and will be beneficial for abiotic stress research. The execution of experiments are satisfactory, however I have some major comments which need to be addressed for the possible publication in Plants Journal.

  1. There are previous studies which demonstrated the role of PIP genes in multiple abiotic stress tolerance, author can mention some points regarding the novelty of the current study in introduction and discuss it.
  2. Since the physiological traits are important to consider for plant vigour, authors are advised to investigate the physiological traits (photosynthesis efficiency, stomatal conductance, and transpiration rate) under drought and salt stress in both transgenic and control plants.  
  3. The figure quality need to be improved especially (Fig.2, Fig. 4a,c)

Author Response

Dear reviewer,

Thank you very much for your advice. We are truly grateful to your critical comments. Based on the comments we received, careful modifications have been made to the R1 manuscript. Please see the attachment, you will find our point-by-point responses to your comments.

I look forward to hearing from you soon.

With best wishes,

Yours sincerely,

Xiaofang Zeng

Reviewer 3 Report

Below are a few suggestions that the authors should consider as they revise the manuscript.

Material and Methods

Page 2

Lines 68-69. Please describe conditions in the glasshouse.

Page 3

Line 109-117. Add plant incubation conditions. Besides that, should have more information regarding soil and temperature.

 Page 4

Line 134. Salt stress treatment lacks all details, such as growing conditions, temperature etc.

Author Response

Dear reviewer,

Thank you very much for your advice. Based on the comments we received, careful modifications have been made to the R1 manuscript. Please see the attachment, you will find our point-by-point responses to your comments.

I look forward to hearing from you soon.

With best wishes,

Yours sincerely,

Xiaofang Zeng

Round 2

Reviewer 1 Report

In the revision of their manuscript, Chen et al address the issues which were raised before, responding to my comments.

Reviewer 2 Report

The revised version looks improved.